# Multi-Class Determination of 64 Illicit Compounds in Dietary Supplements Using Liquid Chromatography–Tandem Mass Spectrometry

**DOI:** 10.3390/molecules25194399

**Published:** 2020-09-24

**Authors:** Dasom Shin, Hui-Seung Kang, Hyungsoo Kim, Guiim Moon

**Affiliations:** New Hazardous Substances Division, Department of Food Safety Evaluation, National Institute of Food and Drug Safety Evaluation, Ministry of Food and Drug Safety, Osong, Cheongju 28159, Korea; shindasom218@korea.kr (D.S.); jungin98@korea.kr (H.K.); luna@korea.kr (G.M.)

**Keywords:** multi-class, illegal compounds, overseas direct purchase, analytical method, LC-MS/MS

## Abstract

In this work, liquid chromatography–tandem mass spectrometry (LC-MS/MS) method was developed and validated for screening and confirmation of 64 illicit compounds in dietary supplements. The target compounds were illegally used pharmaceutical drugs, prohibited compounds, and not authorized ingredients for different therapeutics (sexual enhancement, weight loss, muscular strengthening, and relaxing products). The validation procedure was performed to evaluate selectivity, linearity, limit of detection (LOD), limit of quantification (LOQ), accuracy, and precision according to the Association of Official Analytical Chemists guidelines. The linearity was >0.98 in the range of 0.5–200 µg L^−1^. The LOQs were in the range 1–10 µg kg^−1^ for all target compounds. The accuracy (expressed as recovery) was 78.5–114%. The precision (expressed as the relative standard deviation) was below 9.15%. The developed method was applied for the determination of illicit compounds in dietary supplements collected from websites. As a result, the total detection rate was 13.5% (27 samples detected in 200 samples). The concentrations of detected samples ranged from 0.51 to 226 mg g^−1^. The proposed methodology is suitable for monitoring the adulteration of illicit compounds in dietary supplements.

## 1. Introduction

Consumer demand for dietary supplements that support healthy lifestyles has steadily increased worldwide. In the last decade, illegal adulterants of pharmaceuticals, unapproved drugs, and prohibited ingredients have increased and posed a global challenge in safety control of dietary supplements [1]. The easy access of Internet, overseas direct purchase, along with an increasingly complex global supply chain, has led to the widespread illegal drug distribution worldwide [2]. The abuses of illicit drugs cause adverse and side effects, such as heart attack, psychosis, hypertension, hallucination, tachycardia, agitation, vomiting, seizures, and convulsions [3]. The adulterated illicit compounds in dietary supplements have serious adverse health effect to consumers owing to misuse, overuse, or interaction with other medications [4]. Thus, continuous monitoring studies is the first priority to protect consumer health and safety management of dietary supplements [5].

HPLC equipped with ESI-MS/MS is a well-established and widespread technique for the analysis of various compounds illegally used to adulterants of sexual enhancement and weight loss compounds in dietary supplements [6,7,8]. Due to the increase in the adulteration of illegal compounds in dietary supplements, several analytical methods have been previously reported using liquid chromatography tandem mass spectrometry (LC-MS/MS) [9,10,11,12]. In addition, the multi-class analytical method using LC–MS/MS is widely applied for reliable quantitation and confirmation of illicit compounds [13,14].

In this work, we developed and validated the multi-class determination of 64 illicit compounds in dietary supplements using LC-MS/MS referred to previous studies. The target compounds were a variety of potential chemicals that could be illegally added into dietary supplements including pharmaceutical drugs, prohibited compounds, and not authorized ingredients according to Korean food legislation (Appendix A). To the best of our knowledge, this is the first study to use an analytical method for the simultaneous determination of 64 illicit compounds and to control the custom clearance of illegal substances in Korea. The developed method was applied to real samples of dietary supplements collected by overseas direct purchase. The proposed multi-class methods for the screening and confirmation of compounds have been used for routine monitoring in dietary supplements.

## 2. Results and Discussion

### 2.1. Optimization of LC-MS/MS Conditions

The chromatographic separation was optimized by thoroughly adjusting the LC parameters such as symmetric peaks, column temperature, mobile phase, and flow rate, based on a previous study [5,15,16]. The optimal separation of target compounds was achieved using 0.1% formic acid in aqueous mobile phase and 0.1% formic acid in acetonitrile as organic phase on a BEH C_18_ column (2.1 mm × 150 mm, 3.5 µm, Waters, Dublin, Ireland). As a result, the BEH column offered the best separation, apparent peak shape and signal, and short analysis time (<20 min). Water and acetonitrile showed better performance than methanol for illicit compounds. Formic acid in water was preferred to enhance ionization of target compound during the electrospray process [16,17]. Chromatograms of the 64 compounds obtained from spiked dietary supplement samples with all compounds at the target concentrations are shown in Appendix A. Despite the many kinds of target analytes, the high-quality separation of 64 compounds was achieved. The gradient elution conditions were also optimized by the variation of mobile phases through the repeated analysis of the standard mixture. In addition, eluting the weakly retained compounds, it is necessary to raise the content of aqueous solvent during the gradient profile to increase of the resolution of the column [18,19]. The determination of target compounds was performed using ESI positive and negative ion modes through the direct infusion of standard solutions (100 µg L^−1^). We optimized the characteristic MS/MS parameters specifically for each analyte: capillary voltage of 3.5 kV; desolvation and source temperatures of 500 and 150 °C, respectively; desolvation gas flow of 650 L/h; and cone voltage of 30 V. Collision-induced dissociation was performed using argon. Collision energies (5–60 eV) were tested to give the maximum of intensity for the fragment ions. MRM transition with the most abundant ion was used for quantification, while the other two transitions were used for confirmation ions. In 45 compounds, it was found that singly charged precursor ions [M + H]^+^ were the most abundant, whereas the deprotonated ion [M − H]^−^ was most abundant in 19 compounds (Table 1). MS parameters were optimized in both positive and negative ionization modes with variation of a single setting at a time and evaluation of the sensitivity. Dwell time has a huge influence on the quality of the mass spectra, since lower dwell time leads to more noise on the baseline the peaks. Thus, 16–129 ms of dwell time are desirable for better signal to noise and sensitivity. Due to the large number of substances to be analyzed, a method was developed for simultaneous analysis of 64 illicit compounds using a scheduled multiple reaction monitoring (s-MRM) algorithm monitor. The s-MRM improved the quality of the chromatograms, signal response, and enhanced signal to noise (S/N) ratio, resulting in more data points [20,21].

### 2.2. Method Validation

The LOQ, accuracy, and precision at three target concentrations are shown in Table 2. The linearity of calibration curves was evaluated by six-point calibrations of standards in solvents at the levels in the range of 0.5–200 µg L^−1^. The correlation coefficients (r^2^) were higher than 0.98 in all compounds. Our results show a good linearity that allowed coverage for all target compounds.

The accuracy (expressed as recovery, percent) and precision (expressed as RSD, percent) of target compounds were evaluated in spiked blank samples at three target concentrations (10, 50, and 100 µg L^−1^). The overall recoveries for all target compound ranged 78.5–114%. The precision ranged 0.78–9.15% in intra-day, inter-day, and inter-lab validation processes. These results indicate the good precision and reliability of the proposed method. The LODs and LOQs were calculated by three and ten times the standard deviation divided by the slope of standard curves, respectively. The LOD and LOQ values were <5 µg L^−1^ and <10 µg L^−1^, respectively. Inter-laboratory validation test was conducted to evaluate the ruggedness with three institutions. As a result, recovery ranged 61.1–132% with RSD less than 15% for all target compounds. The analytical method showed an acceptable accuracy and precision, with satisfactory values for all method validation parameters according to the requirements of the AOAC guidelines.

Most of collected samples were capsules, powders, tablets, and soft-gels (Appendix A). However, herbal supplements may have a complex matrix that can potentially interfere with the determination of target compounds and cause matrix effect in mass analysis [10]. The matrix effect should be evaluated for further method development and for a variety of plant food and herbal supplement samples.

### 2.3. Application of Real Samples

The proposed method was applied in dietary supplement samples (*n* = 200) randomly collected from 10 websites. The product category (indication for which product was marketed) of samples was classified into sexual performance (15.5%), weight loss (47%), muscle strengthening enhancement (19%), relaxation (5.5%), and others (13%). The type of samples was categorized into capsules (51%), powders (25%), tablets (10.5%), soft-gels (6%), liquids (5.5%), tea bag (0.5%), and energy bars (1%) (Appendix A). The detection numbers and concentrations in collected dietary supplements are described in Table 3. The total detection rate was 13.5% (27 out of 200 samples). Eleven kinds of compounds were detected in 27 samples. In weight loss products, 16 samples were detected in 94 samples (17%). For muscular strengthening products, seven samples were detected in 38 samples (18.4%). In relaxing products, six samples were detected in 11 samples (54.5%). For sexual enhancement products, four samples detected in 31 samples (12.9%). Moreover, the combination of two or more compounds in one sample was detected in four samples. Examples of chromatogram and mass spectra are shown in Appendix A.

Synephrine was the most frequently detected in dietary supplements (eight samples), and the concentrations ranged 3.70–75.7 mg g^−1^. Synephrine is a substance that occurs naturally in plants and is added to dietary supplements for weight-loss and sports performance [22]. However, synephrine is prohibited because it may have adverse effect on human health. Synephrine has structural and pharmacological similarities with ephedrine; therefore, it can cause similar side effects. Furthermore, synephrine and DHEA were added to the inspection program in competitions of the World Anti-Doping Agency [22].

Yohimbine was detected in six samples at concentrations of 0.51–11.0 mg g^−1^. A similar concentration (6.6 mg g^−1^) of yohimbine was detected in one sample [23]. In Korea, yohimbine is a drug used in veterinary medicine in livestock products based on maximum residue limit [24]. Although drugs of yohimbine are no longer prescribed by US physicians, yohimbine remains an ingredient in dietary supplements for sexual and sports enhancements. Dietary supplements containing yohimbine can pose a significant risk to consumers [25]. Yohimbine is a natural tryptamine alkaloid extracted from the bark of various plants. Due to the side effects, this substance in sexual enhancement products carries a psychological health risk [3].

5-Hydroxytryptophan (5-HTP) known as oxitriptan was detected in the range of 3.50–226 mg g^−1^ in six samples. 5-HTP is a naturally occurring amino acid and chemical precursor as well as a metabolic intermediate in serotonin synthesis. In certain countries, 5-HTP is added in dietary supplements to treat depression and insomnia [26]. Melatonin was detected in two samples (3.35–6.74 mg g^−1^). Melatonin has been sold as on over-the-counter drug and is not regulated by the Food and Drug Administration, whereas, in Europe and Asia-Pacific, it is a prescription drug. 5-HTP and melatonin may cause side effects such as delirium, risk of falling down, and sleep-related eating disorder.

Cascarosides were detected in three samples (5.77–9.62 mg g^−1^). Sennosides were detected in two samples (13.9–35.0 mg g^−1^). In a previous study, sennoside A and B were detected in the range of 0.41–3.27 mg g^−1^ in 16 samples. Cascara and senna extracts are well known effective laxative drugs and used in the treatment of intestinal constipation. Some side effects have been reported in the administration of senna extracts [27]. Icariin was detected in two samples (0.77–3.03 mg g^−1^). In a previous study, icariin was detected in 19 samples at 0.1–19.0 mg g^−1^ in adulterated products [23]. Kavain (18.8 mg g^−1^) was detected in one sample. These findings suggest that plant origin dietary supplements are commonly used to promote sexual health and weight loss. Many herbal supplements have potential benefits, however the health risks of various phytotherapeutic compounds need to be elucidated.

Overall, illicit compounds continue to be detected in dietary supplements for sexual enhancement and weight loss. The pharmaceuticals in dietary supplements have potential health risks due to misuse, over use, or interaction with other pharmaceuticals in drugs and supplements. The adverse effects reported for the use of dietary supplements include stroke, acute liver injury, kidney failure, pulmonary embolisms, and death [4]. Further studies are needed to improve the wide coverage of sampling including websites and more private ways of social networking.

## 3. Materials and Methods

### 3.1. Chemical and Reagents

Psychoactive substances (amphetamine, 99.9%; atropine, 97.0% BMPEA, 95.0%; fenfluramine hydrochloride, 99.2%; lorcaserin hydrochloride, 99.0%; phendimetrazine tartrate, 99.9%; phentermine hydrochloride, 99.9%; and salicylic acid, 99.9%) were obtained from the Ministry of Food and Drug Safety (Cheongju, South Korea) in accordance with Narcotics Control Act. In addition, synthetic compounds (7-keto-DHEA, 99.9%; cascaroside A and B, 98.0%; cascaroside C, 97%; cascaroside, D 99%; cimifugin, 99.0%; icaritin, 99.0%; n-nitrosofenfluramine, 99.0%; salicin, 99.0%; and synephrine, 99%) were synthesized by the Ministry of Food and Drug Safety (Cheongju, South Korea) for testing. Other compounds of high purity grade (≥95%) were purchased from Wako Pure Chemical Industries Inc. (Osaka, Japan), USP (Rockville, MD, USA), or Sigma-Aldrich (Buchs, Switzerland). The manufacturer and purity for 64 chemical standards are presented in Appendix A. All reagents and solvents (analytical grade) were purchased from Merck Inc. (Darmstadt, Germany) or Sigma-Aldrich (Buchs, Switzerland). The samples were purified by filtration through polytetrafluoroethylene (PTFE) filter provided by Teknokroma (Barcelona, Spain). Individual standard stock solution was prepared by dissolving each standard in methanol with a concentration of 1000 µg mL^−1^ and were stored at −4 °C. Working mix standard solutions containing all compounds were daily prepared by appropriate dilution of the stock standard solution with 70% methanol.

### 3.2. Sample Preparation

The dietary supplements in the form of capsules, tablets, powder, oral liquids, and soft-gels were obtained through overseas direct purchase. All tablets were homogenized with a blender. The shells of all of the capsules and soft-gels were removed to mix the powder inside. First, 1.0 ± 0.01 g sample was transferred into a 50-mL volumetric flask. Each sample (1 g) was accurately weighed, and then mixed in water (15 mL) for 1 min. After that, methanol was added (20 mL), and the samples sonicated for 20 min to extract the analytes. Methanol was added into the final volume up to 50 mL. The sample extracts were filtered through a polytetrafluoroethylene (PTFE) syringe filter with 0.22-μm pore size before LC-MS/MS analysis.

### 3.3. LC-MS/MS Analysis

LC-MS/MS analysis was performed on a Waters Acquity I-Class UPLC system (Waters, Milford, MA, USA) coupled with a Xevo TQ-S (Triple Quadrupole Mass Spectrometer) equipped with an electrospray ionization source (Waters, Milford, MA, USA). The chromatographic separation was carried out using Acquity BEH C_18_ column (2.1 mm × 150 mm, 3.5 µm, Waters, Dublin, Ireland). The mobile phase was made up of Solvent A (0.1% formic acid in water), and Solvent B (0.1% formic acid in acetonitrile). The oven and column temperature were maintained at 40 °C, and the flow rate was set at 0.3 mL min^−1^. The optimized gradient was as follows: 0–3 min, 5% B; 3–18 min, 80% B; 18–18.1 min, 100% B; 18.1–20.9 min, 100% B; 20.9–21 min 5% B; and 21–25 min, 5%. The injection volume was 5 µL. Mass spectrometer conditions were set as follows: the capillary voltage, 3.5 kV in ESI positive mode and −2.8 kV in the ESI negative mode. The desolvation and source temperatures were set at 500 and 150 °C, respectively. The cone and desolvation gas flow rates (nitrogen) were 60 and 600 L h^−1^, respectively. Collision-induced dissociation was performed using argon as the collision gas at a pressure of 4 × 10^−3^ mbar in the collision cell. Data collection was implemented in multiple reaction monitoring (MRM) mode using the MassLynx software (Waters, UK). The [M + H]^+^ and [M–H]^–^ ions were selected as precursor ions; the two or three most intense product ions were used as product ions. Table 1 describes the optimized mass parameters, MRM transition, and retention times of the 64 illicit compounds.

### 3.4. Method Validation

The method validation procedure was performed through estimation of the linearity, limit of detection (LOD), limit of quantitation (LOQ), accuracy, and precision based on AOAC guideline (AOAC, 2016). Calibration curves were obtained by linear regression analysis over the target concentration levels in the range of 0.5–200 µg L^−1^, depending on the compounds. The LODs and LOQs were calculated by three and ten times the standard deviation divided by the slope of standard curves, respectively. The accuracy (recovery, percent) and precision (relative standard deviation, percent) were determined through the analysis of negative sample of dietary supplements spiked at 10, 50, and 100 µg kg^−1^ (bisacodyl and yohimbine were tested at 1, 5, and 10 µg kg^−1^) in five replicates for 64 illicit compounds within a single day. The intra- and inter-day precision were also determined in triplicate within one day and over three consecutive days, respectively.

## 4. Conclusions

In this study, we reported the development and optimization multi-class determination of 64 selected compounds in dietary supplements using LC-MS/MS. The present method showed suitable linearity, accuracy, precision, LOD, and LOQ to detect the multiple classes of 64 illicit compounds in dietary supplements. The detection rate was 13.5% (27 out of 200 samples). Eleven compounds were detected in 27 dietary supplement samples. Synephrine and yohimbine were the most detected compounds in weight loss and muscular strengthening products. The proposed method was successfully validated and applied for identification and confirmation of illicit compounds in dietary supplements. As the dietary supplement industry continues to grow in the world, adulterated products should be regularly monitored to protect public from potential health risks.

## Figures and Tables

**Table 1 molecules-25-04399-t001:** MRM transition and optimized parameters of LC-MS/MS for 64 target compounds.

Compounds	Ion Mode (ESI +/−)	Molecular Weight (g/mol)	Precursor Ion (*m*/*z*)	Product Ion (*m*/*z*)	Collision Energy (eV)	Cone Voltage (V)	Retention Time (min)
2,4-Dinitrophenol (2,4-DNP)	−	184.1	183.1	**109**	25	30	9.46
				137	20		
				153	15		
7-keto-dehydroepiandrosterone (7-keto-DHEA)	+	302.4	303.3	**81**	24	30	9.72
				267	15		
				285	15		
Amphetamine	+	135.2	136.1	41	26	20	4.98
				**91**	15		
				119	10		
Asarone	+	208.3	209.1	151	24	30	13.3
				179	22		
				**194**	16		
Atropine	+	289.4	290.1	77	35	20	6.37
				93	28		
				**124**	22		
Berberine	+	336.4	336.2	**292**	28	25	8.63
				306	28		
				320	25		
beta-methylphenethylamine (BMPEA)	+	135.2	136.1	**91**	15	30	4.79
				119	10		
				-	-		
Bisacodyl	+	361.4	362.1	167	50	30	10.9
				**184**	28		
				226	18		
Buformin	+	157.1	158.2	43	20	30	1.41
				**60**	15		
				116	15		
Cascaroside A	−	580.1	579.1	268	52	35	5.89
				**297**	38		
				459	20		
Cascaroside B	−	580.1	579.1	268	52	35	5.31
				297	38		
				459	20		
Cascaroside C	−	563.8	563.2	251	64	30	7.23
				**281**	40		
				443	24		
Cascaroside D	−	563.8	563.2	251	64	30	6.96
				**281**	40		
				443	24		
Chlorothiazide	−	295.7	293.9	115	52	30	3.99
				179	46		
				**214**	30		
Cimifugin	+	306.3	307.2	221	28	35	7.54
				**235**	28		
				259	28		
Dehydroepiandrosterone (DHEA)	+	288.4	289.2	197	18	30	12.5
				213	18		
				**253**	12		
Echinacoside	−	786.7	785.3	133	80	40	6.53
				**161**	46		
				623	36		
Ephedrine	+	165.2	166.2	117	20	35	3.51
				133	20		
				**148**	13		
Fenfluramine	+	231.3	232.2	109	35	35	8.33
				**159**	15		
				187	15		
Fluoxetine	+	309.3	310.1	44	10	30	10.6
				117	50		
				**148**	10		
Glibenclamide	−	494.0	492.1	127	50	30	13.9
				**170**	32		
				367	18		
Gliclazide	−	323.4	322.1	80	48	40	12.45
				106	42		
				**170**	22		
Glimepiride	−	490.6	489.3	**225**	34	45	14.3
				350	20		
				364	28		
Glipizide	−	445.5	444.1	154	54	30	11.4
				**170**	34		
				319	20		
Hesperidin	+	610.2	611.2	153	52	30	8.02
				**303**	22		
				449	12		
Hydrastine	+	383.4	384.2	190	20	35	7.32
				293	25		
				**323**	20		
Hydrochlorothiazide	−	297.7	295.9	78	28	40	4.80
				205	24		
				**269**	18		
Hydroflumethiazide	−	331.3	329.9	160	38	30	6.62
				**239**	24		
				303	20		
Icariin	+	676.7	677.3	313	58	30	9.51
				**369**	32		
				531	16		
Icaritin	+	368.4	369.1	135	35	30	15.8
				187	40		
				**313**	23		
Kavain	+	230.3	231.1	**115**	14	35	11.9
				153	22		
				185	15		
Levodopa	+	197.2	198.1	107	25	20	0.82
				**152**	15		
				181	10		
Levothyroxine	+	776.9	777.6	324	54	25	10.7
				**351**	46		
				605	40		
Liothyronine	+	651.0	651.7	**197**	68	30	9.94
				225	42		
				479	34		
Lorcaserine	+	195.7	196.1	**129**	28	30	7.61
				139	22		
				144	20		
Lovastatin	+	404.5	405.3	**199**	12	35	16.1
				225	14		
				285	12		
Magnoflorine	+	342.4	342.2	192	35	30	6.13
				**265**	25		
				298	25		
Melatonin	+	232.3	233.1	130	40	30	8.11
				143	25		
				**174**	16		
Methylchlorothiazide	-	360.2	357.9	194	22	30	9.25
				258	18		
				**322**	14		
Mexamine (5-methoxytryptamine1)	+	190.2	191.1	130	38	30	5.29
				**159**	22		
				174	16		
N-nitrosofenfluramine	+	260.3	261.1	109	44	30	14.1
				**159**	22		
				187	12		
Noopept	+	318.4	319.2	**70**	22	35	9.20
				188	14		
				216	10		
Oxilofrine	+	181.2	182.1	105	22	40	0.83
				133	18		
				**149**	20		
Oxindole	+	133.2	134.1	51	35	45	6.87
				79	22		
				**106**	16		
Oxitriptan (5-hydroxytryptohpane)	+	220.2	221.1	106	32	30	0.83
				**134**	24		
				160	24		
Phendimetrazine	+	191.3	192.2	74	18	30	5.26
				117	21		
				**146**	24		
Phenformin	+	205.3	206.1	**60**	20	35	4.90
				77	35		
				164	20		
Phenolphtalein	+	318.3	319.1	141	42	25	10.7
				197	30		
				**225**	22		
Phentermin	+	149.2	149.9	55	18	30	5.86
				65	34		
				**133**	10		
Picamilon	+	208.2	209.1	78	25	30	1.77
				**106**	20		
				108	22		
Rauwolscine (α-yohimbine)	+	354.4	355.2	117	40	30	7.49
				**144**	32		
				212	22		
Reserpine	+	608.7	609.3	174	35	30	10.6
				**195**	35		
				397	30		
Salbutamol	+	239.3	240.1	121	25	35	2.28
				**148**	20		
				166	15		
Salicin	−	286.3	331.1	93	50	40	3.94
				**123**	12		
				285	10		
Salicylic acid	-	138.1	136.9	**65**	26	45	8.44
				93	25		
				-	-		
Scopolamine	+	303.4	304.2	121	20	35	5.31
				**138**	15		
				156	15		
Sennoside A	−	862.8	861.2	224	40	30	7.55
				**386**	36		
				699	28		
Sennoside B	−	862.8	861.2	224	40	30	7.01
				**386**	36		
				699	28		
Serotonin (5-hydroxytryptamine)	+	176.2	177.1	105	26	50	1.43
				**115**	24		
				142	22		
Synephrine	+	167.2	168.1	107	26	30	0.82
				119	16		
				**135**	20		
Tolbutamide	+	270.4	271.1	74	12	55	11.5
				**91**	30		
				155	16		
Trichloromethiazide	−	380.7	377.9	215	34	35	8.80
				**242**	22		
				306	14		
				95	18		
				-	-		
Vinpocetine	+	350.5	351.1	**280**	26	30	9.98
				294	25		
				308	25		
Yohimbine (β-yohimbine)	+	354.4	355.2	117	40	35	7.49
				**144**	32		
				212	22		

The bold text expresses the quantification ion.

**Table 2 molecules-25-04399-t002:** LOQ, accuracy, and precision at three target levels in dietary supplement.

Compounds	Linearity (r^2^)	LOD (ng kg^−1^)	LOQ (ng kg^−1^)	Target Concentrations (ng kg^−1^)	Accuracy (% Recovery)		Precision (% RSD)	
					Intra-Day	Inter-Day	Intra-Day	Inter-Day
2,4-Dinitrophenol (2,4-DNP)	0.9965	2	6	10	91.4	92.7	5.41	2.67
				50	107	102	0.83	4.01
				100	96.4	100	1.68	3.82
7-keto-dehydroepiandrosterone (7-keto-DHEA)	0.9977	1	2	10	86.9	103	2.26	3.13
				50	108	100	2.17	3.45
				100	107	99.8	0.81	3.27
Amphetamine	0.9995	1	4	10	109	95.6	2.48	4.66
				50	104	99.7	2.19	2.53
				100	104	100	1.54	2.91
Asarone	0.9991	1	4	10	89.8	94.6	6.59	5.02
				50	92.0	102	6.95	3.85
				100	88.1	99.4	6.17	3.68
Atropine	0.9993	2	6	10	99.2	92.8	4.50	5.72
				50	109	92.3	5.45	4.54
				100	106	98.5	0.80	2.67
Berberine	0.9988	2	6	10	84.6	91.2	3.65	3.34
				50	104	99.9	2.32	4.78
				100	102	102	3.23	3.57
beta-methylphenethylamine (BMPEA)	0.9998	1	3	10	111	95.2	5.04	3.19
				50	105	100	3.21	2.20
				100	102	100	2.92	1.86
Bisacodyl	0.9978	0.3	1	10	89.2	101	1.76	1.32
				50	96.5	92.9	0.94	2.05
				100	98.9	92.3	1.92	1.62
Buformin	0.9965	2	6	10	87.4	86.4	1.58	4.94
				50	111	103	2.20	2.57
				100	114	97.9	1.29	2.76
Cascaroside A, B, C, D	0.9991	3	9	10	86.9	100	8.02	3.69
				50	100	101	1.13	3.33
				100	100	98.9	1.35	2.64
Chlorothiazide	0.9962	3	8	10	95.1	88.6	2.68	4.52
				50	102	102	2.14	4.86
				100	103	102	1.51	2.12
Cimifugin	0.9994	1	4	10	94.1	88.9	3.16	6.29
				50	102	102	2.83	4.74
				100	102	96.6	4.22	2.58
Dehydroepiandrosterone (DHEA)	0.9997	1	5	10	95.2	99.3	6.90	5.86
				50	101	97.1	6.12	3.88
				100	89.1	98.1	2.57	5.75
Echinacoside	0.9993	3	9	10	101	95.8	7.55	6.73
				50	94.5	96	7.33	6.39
				100	93.9	103	3.53	4.37
Ephedrine	0.9975	1	4	10	96.6	90.5	1.33	3.24
				50	103	99.3	1.64	1.81
				100	106	100	1.34	2.62
Fenfluramine	0.9984	2	6	10	91.8	91.1	2.69	4.68
				50	104	105	8.25	3.18
				100	103	101	3.72	2.75
Fluoxetine	0.9992	1	5	10	95.2	91.4	5.55	2.46
				50	114	102	1.57	2.32
				100	110	103	1.35	1.98
Glibenclamide	0.9994	2	6	10	89.0	95.9	3.63	2.75
				50	104	97.3	1.95	1.28
				100	103	96.6	1.15	2.59
Gliclazide	0.9994	1	3	10	84.4	98.5	2.25	2.78
				50	101	97.8	0.98	2.79
				100	99.8	96.2	1.39	2.45
Glimepiride	0.9997	1	2	10	89.3	99.3	1.54	1.88
				50	102	100	0.98	1.77
				100	101	98.3	1.65	2.12
Glipizide	0.9991	1	4	10	80.1	94.7	1.09	3.17
				50	101	98.7	1.26	2.27
				100	101	96.8	1.35	3.05
Hesperidin	0.9999	1	3	10	97.5	97.6	4.01	5.31
				50	104	100	1.34	3.33
				100	103	101	2.43	1.74
Hydrastine	0.9980	1	4	10	91.3	87.3	8.54	4.63
				50	107	96.9	4.00	4.23
				100	106	99.8	4.18	3.25
Hydrochlorothiazide	0.9989	2	8	10	95.4	98.8	7.53	5.47
				50	98.9	97.5	4.37	2.89
				100	99.7	100	6.67	1.58
Hydroflumethiazide	0.9989	2	7	10	97.4	92.5	6.21	7.34
				50	107	92.7	6.09	4.66
				100	112	94.7	6.51	2.23
Icariin	0.9984	1	4	10	85.1	92	1.68	2.01
				50	105	98.9	1.12	2.64
				100	106	100	1.39	2.57
Icaritin	0.9985	1	3	10	99.3	89.3	5.97	2.58
				50	106	102	2.81	3.31
				100	101	102	1.77	1.61
Kavain	0.9994	1	3	10	84.1	85	2.15	3.02
				50	104	103	2.07	1.32
				100	99.5	100	1.32	1.09
Levodopa	0.9996	1	5	10	86.6	92.6	2.74	4.85
				50	98.8	102	1.17	2.41
				100	97.2	103	1.55	1.18
Levothyroxine	0.9997	1	5	10	96.1	100	3.97	3.33
				50	101	100	2.63	3.70
				100	103	98.8	4.33	2.31
Liothyronine	0.9999	2	5	10	96.9	95	2.50	3.15
				50	104	98.3	0.99	4.30
				100	104	98.9	2.16	2.25
Lorcaserine	0.9979	1	4	10	85.0	92.9	2.70	3.93
				50	105	102	3.75	3.41
				100	107	103	1.98	3.17
Lovastatin	0.9989	1	3	10	88.4	91.5	2.22	2.50
				50	105	108	3.60	1.51
				100	98.7	108	3.96	0.93
Magnoflorine	0.9985	2	8	10	99.3	94.4	5.97	6.03
				50	106	107	2.81	2.92
				100	101	102	1.77	2.61
Melatonin	0.9994	1	2	10	91.3	81.9	3.71	4.23
				50	104	102	1.61	2.36
				100	99.2	100	2.13	2.46
Methylchlorothiazide	0.9981	2	8	10	81.2	96.8	5.59	6.42
				50	103	99.1	3.77	5.84
				100	100	99.2	1.93	3.73
Mexamine (5-methoxytryptamine1)	0.9960	1	5	10	95.7	99.4	9.15	6.53
				50	112	102	4.23	4.75
				100	108	105	1.72	3.85
N-nitrosofenfluramine	0.9989	1	5	10	83.1	88	2.89	3.15
				50	105	101	2.51	2.63
				100	103	99	1.03	2.78
Noopept	0.9996	1	4	10	83.9	85.6	2.32	3.03
				50	101	98.3	1.66	4.59
				100	101	98.1	1.72	4.31
Oxilofrine	0.9988	1	5	10	97.4	93	2.15	3.29
				50	111	106	5.04	2.67
				100	112	98.1	3.11	1.98
Oxindole	0.9997	1	2	10	102	90	2.76	2.96
				50	102	97.9	2.10	3.41
				100	103	99.3	1.34	1.64
Oxitriptan (5-hydroxytryptohpane)	0.9991	2	8	10	97.2	85.9	2.83	4.69
				50	102	98	2.56	3.33
				100	105	100	2.99	1.77
Phendimetrazine	0.9987	1	5	10	91.0	89.4	2.37	4.90
				50	103	95.8	2.64	3.06
				100	100	99.1	2.66	1.93
Phenformin	0.9976	1	4	10	88.6	83	7.17	7.18
				50	110	98.1	5.41	3.52
				100	106	101	3.57	3.52
Phenolphtalein	0.9993	1	5	10	90.4	86.2	3.09	3.42
				50	103	100	1.21	2.12
				100	99.0	99.3	3.51	2.95
Phentermin	0.9992	1	4	10	101	85.9	3.02	4.84
				50	104	100	1.30	3.28
				100	105	100	2.50	3.11
Picamilon	0.9992	1	4	10	98.9	85.5	1.99	4.03
				50	102	97.5	1.23	2.55
				100	103	97.5	0.78	1.84
Rauwolscine (α-yohimbine)	0.9989	0.3	1	10	103	91.9	4.08	5.98
				50	102	96.6	5.00	3.15
				100	91.6	99.1	4.57	4.34
Reserpine	0.9997	1	5	10	99.2	97.5	2.14	2.10
				50	102	100	2.13	3.23
				100	102	99.1	1.67	3.18
Salbutamol	0.9981	2	6	10	87.3	85.5	1.97	3.18
				50	107	100	2.42	2.61
				100	109	100	1.05	1.25
Salicin	0.9994	5	10	10	97.8	92	5.41	4.61
				50	100	95.4	1.91	4.72
				100	102	94.1	4.49	4.24
Salicylic acid	0.9987	3	8	10	90.3	98.1	8.22	4.85
				50	98.1	95.5	4.50	4.97
				100	97.2	98.8	4.82	6.85
Scopolamine	0.9966	3	6	10	92.0	87.5	2.80	8.09
				50	110	99.9	4.76	4.17
				100	108	101	4.68	2.47
Sennoside A, B	0.9943	3	8	10	78.5	98.9	4.83	3.74
				50	98.4	100	4.69	5.00
				100	102	98.9	4.58	4.54
Serotonin (5-hydroxytryptamine)	0.9937	3	9	10	88.0	86.3	6.02	6.58
				50	108	101	5.32	4.40
				100	108	104	2.39	2.78
Synephrine	0.9970	3	6	10	92.5	94.6	6.14	7.59
				50	112	102	3.97	3.48
				100	95.6	93.6	3.44	3.34
Tolbutamide	0.9997	1	4	10	92.3	94.1	2.90	2.23
				50	103	100	2.30	1.83
				100	103	100	1.92	1.49
Trichloromethiazide	0.9973	3	8	10	83.1	91.5	5.15	7.52
				50	104	97.4	7.41	2.38
				100	102	96.3	2.80	1.92
Vinpocetine	0.9994	3	3	10	92.0	87.9	2.17	3.85
				50	110	100	2.10	2.10
				100	109	101	1.00	3.03
Yohimbine (β-yohimbine)	0.9989	0.3	1	10	103	91.9	4.08	5.98
				50	102	96.7	5.00	3.15
				100	91.6	99.2	4.57	4.34

**Table 3 molecules-25-04399-t003:** Detected numbers and concentrations in dietary supplements by product category.

Product Category by Labelling (Sample Number)	Compounds	Detected Number	Detected Sample No.	Concentrations		Contents
				Mean (mg g^−1^)	Range (mg g^−1^)	Contents per Unit (mg/Unit)
Sexual enhancement products (*n* = 31)	DHEA	1	S-3	39.7		21.4
	Icariin	2	S-2^1)^, S-88	1.90	0.77–3.03	1.34–2.12
	Magnoflorine	1	S-2^1)^	0.54		0.38
Weight loss products (*n* = 94)	Cascarosides	3	S-4, S-77, S-79	8.02	5.77–9.62	1.47–6.16
	Melatonin	1	S-171	3.35		4.02
	Mexamine	1	S-7^1)^	3.64		4.14
	Oxitriptan (5-HTP)	2	S-7^1)^, S-172	20.4	3.50–37.2	3.99–23.1
	Sennosides	2	S-161, S-162	24.5	13.9–35.0	9.89–19.6
	Synephrine	5	S-6, S-7^1)^, S-10, S-13, S-140	27.2	3.70–75.7	2.92–67.4
	Yohimbine	2	S-7^1)^, S-60	2.32	1.69–2.94	1.03–1.47
Muscular strengthening products (*n* = 38)	Synephrine	3	S-61, S-107, S-155^1)^	33.3	14.9–56.5	18.3–134
	Yohimbine	4	S-9, S-16, S-142, S-155^1)^	4.60	0.51–11.0	1.78–9.31
Relaxing products (*n* = 11)	Kavain	1	S-73	18.8	18.8	14.3
	Melatonin	1	S-48^1)^	6.74		1.94
	Oxitriptan (5-HTP)	4	S-17, S-22, S-48^1)^, S-82	115	56.6–226	20.6–140
Others (*n* = 26)						
Total (*n* = 200)	11 compounds	33	27 samples		0.51–226	0.38–140

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
