# Peer review of "Multi-Class Determination of 64 Illicit Compounds in Dietary Supplements Using Liquid Chromatography–Tandem Mass Spectrometry"

_molecules, 2020, doi:10.3390/molecules25194399_

Round 1

Reviewer 1 Report

/c/v

Comments on the manuscript molecules-Manuscript ID: molecules-913022

Title: Multi-class determination of 66 illicit compounds in dietary
supplements using liquid chromatography–tandem mass spectrometry

  1. Abstract: 0.5-200 mg Kg-1 is different from 0.5-200 mgL-1 noted in 2.2 Method, page 3.
  2. Please explain the difference of retention time data for compounds in Figure 1 and Table 1. For example: r.t. is 9.46 for 2.4-Dinitrophenol in Figure 1, but r.t is 9.28 for 2.4-Dinitrophenol in Table 1.
  3. Figure 1 can be considered to remove to Supplement part, if the information of Figure 1 are included in Table 1.
  4. Please explain why Sennoside A and B had the same retention time data under different condition, e.g. 8.68 e4 and 8.24 e 4.
  5. Page 4: cascaroside C 97%; cascaroside D 99%; if their purity are so high, why Figure 1 showed two peaks of retention time for cascaroside C D and the ratio of the two peaks seems unreasonable.
  6. Why Cascarosid A and B had very high purity and the same molecule weight, whereas they showed two peaks and their retention times are different?
  7. The descriptions of many data in text could be described more detail.

Author Response

Comments on the manuscript molecules-Manuscript ID: molecules-913022

Title: Multi-class determination of 66 illicit compounds in dietary supplements using liquid chromatography–tandem mass spectrometry

Authors: Thank you very much for your comments. We have thoroughly revised article based on the reviewer’s suggestion.

Abstract: 0.5-200 mg Kg-1 is different from 0.5-200 mgL-1 noted in 2.2 Method, page 3.

Authors: We have change the unit to mg L-1

Please explain the difference of retention time data for compounds in Figure 1 and Table 1. For example: r.t. is 9.46 for 2.4-Dinitrophenol in Figure 1, but r.t is 9.28 for 2.4-Dinitrophenol in Table 1.

Authors: We have confirmed the retention time (RT) at 9.46 for 2.4-Dinitrophenol. The retention time for all compounds were confirmed and corrected in figure and tables.

Figure 1 can be considered to remove to Supplement part, if the information of Figure 1 are included in Table 1.

Authors: As you suggested, Figure 1 was moved to supplement part.

Please explain why Sennoside A and B had the same retention time data under different condition, e.g. 8.68 e4 and 8.24 e4.

Authors: We have revised Figure 1. Each chromatogram was separately presented. Sennoside A and B were stearic isomer that have similar structure. Thus, the retention time is different under same condition.

Page 4: cascaroside C 97%; cascaroside D 99%; if their purity are so high, why Figure 1 showed two peaks of retention time for cascaroside C D and the ratio of the two peaks seems unreasonable.

Authors: We have revised Figure 1. The intensity of chromatograms for two compounds can be acceptable.

Why Cascarosid A and B had very high purity and the same molecule weight, whereas they showed two peaks and their retention times are different?

Authors: We have revised Figure 1. Cascaroside A and B have similar structure. Thus, the retention time is different under same condition.

The descriptions of many data in text could be described more detail.

Authors: All of manuscript was revised based on reviewer’s suggestion.

Reviewer 2 Report

The paper by Shin et al. describes an LC-MS/MS method for the detection of illegal compounds (drugs) in food supplements and similar. The authors should deeply check the English since many mistakes are present in the manuscript.

In Figure 1 Cascaroside AB and Cascaroside CD are reported twice, the compounds levodopa, oxilofrine, oxitripan 5-HTP, and serotonin 5HT elute all at 0.83. Please report the mass spectra and tandem MS spectra of these compounds in supporting information. The Metformin elutes close to dead volume.

Please correct the decimal digits of Inter-day column in table 2. Why do you calculate the errors only for Inter-day values? Add a column with LOD values and one with linearity.

Table 3. Detected number, concentrations in dietary supplements by product category. Please rewrite

Please better elucidate the adverse effects for consumers when ingesting the analysed products.

Author Response

The paper by Shin et al. describes an LC-MS/MS method for the detection of illegal compounds (drugs) in food supplements and similar. The authors should deeply check the English since many mistakes are present in the manuscript.

Authors: Thank you very much for your comments. We have thoroughly revised article based on the reviewer’s suggestion. The grammar was deeply checked.

In Figure 1 Cascaroside AB and Cascaroside CD are reported twice, the compounds levodopa, oxilofrine, oxitripan 5-HTP, and serotonin 5HT elute all at 0.83. Please report the mass spectra and tandem MS spectra of these compounds in supporting information. The Metformin elutes close to dead volume.

Authors: Thank you for your comments. We have revised Figure 1 and chromatograms of cascaroside A B C D were corrected. Based on the inter-laboratory validation results, data for metformin and trigonelline was deleted. The inter–laboratory validation results for other compounds was added in manuscript as follows:

Inter-laboratory validation test was conducted to evaluate the ruggedness with three institutions. As a result, recovery was ranged from 61.1 to 132% with RSD less than 15% for all target compounds.

Please correct the decimal digits of Inter-day column in table 2. Why do you calculate the errors only for Inter-day values? Add a column with LOD values and one with linearity.

Authors: Based on your comments, Table 2 was revised and added LODs and linearity. The errors for inter-day values were deleted because it is not meaningful.

Table 3. Detected number, concentrations in dietary supplements by product category. Please rewrite

Authors: Table 3 was corrected. The sample number, detected sample no. and the summary were added.

Please better elucidate the adverse effects for consumers when ingesting the analysed products.

Authors: We have revised article as follows:

Overall, the illicit compounds continue to detect in dietary supplements for sexual enhancement and weight loss. The pharmaceuticals in dietary supplements have the potential health risk due to misuse, over use, or interaction with other pharmaceuticals in drug and supplements. The adverse effect was reported for use of dietary supplements include stroke, acute liver injury, kidney failure, pulmonary embolisms, and death [4]. Further studies are needed to improve the wide coverage of the sampling including websites and more private ways of social networking.

Reviewer 3 Report

In this manuscript, the authors reported the development, validation and application a LC-MS-MS methodology for the determination of 66 illicit compounds in different dietary supplements. The method parameters were adequately reported. Validation details, quantitation results on 200 dietary supplements were clearly summarized. Overall the reported LC-MS-MS method has valuable potential in monitoring the illicit components in dietary supplements.

Main suggestions:

  1. Introduction can be more informative. The current methodologies for similar analysis should be added. Comparing to these methods, what’s the advantage or novelty of the current reported method?
  2. Fig. 1 can be moved into supplementary data.
  3. Consider adding a summary table on the quantitative results of the 27 samples that contained detected illicit compounds.
  4. The potential matrix effect of different forms of dietary supplements was not assessed or discussed. The specificity of the proposed method in different forms of dietary supplement samples need to be verified. Consider adding representative MS chromatograms on analyzed dietary samples to show the peak detection quality.

Author Response

In this manuscript, the authors reported the development, validation and application a LC-MS-MS methodology for the determination of 66 illicit compounds in different dietary supplements. The method parameters were adequately reported. Validation details, quantitation results on 200 dietary supplements were clearly summarized. Overall the reported LC-MS-MS method has valuable potential in monitoring the illicit components in dietary supplements.

Authors: Thank you very much for your comments. We have thoroughly revised article based on the reviewer’s suggestion.

Main suggestions:

Introduction can be more informative. The current methodologies for similar analysis should be added. Comparing to these methods, what’s the advantage or novelty of the current reported method?

Authors: Thank you very much for your comments. We have thoroughly revised article based on the reviewer’s suggestion.

In this work, we developed and validated the multi-class determination of 64 illicit compounds in dietary supplements using LC-MS/MS refer to previous studies. The target compounds were a variety of potential chemicals that could be illegally added into dietary supplements including pharmaceutical drugs, prohibited compounds, and not authorized ingredients according to Korean food legislation (Table S1). To the best of our knowledge, this study is the first study to use analytical method for the simultaneous determination of 64 illicit compounds and to control the custom clearance of illegal substances in Korea. The developed method was applied to real samples of dietary supplements collected by overseas direct purchase. The proposed multi-class methods for the screening and confirmation of compounds have been used for routine monitoring in dietary supplements.

Fig. 1 can be moved into supplementary data.

Authors: Figure 1 was moved to supplement part.

Consider adding a summary table on the quantitative results of the 27 samples that contained detected illicit compounds.

Authors: We have corrected Table 3 as your recommended.

The potential matrix effect of different forms of dietary supplements was not assessed or discussed. The specificity of the proposed method in different forms of dietary supplement samples need to be verified.

Authors: We have added the discussion as follows:

Most of collected samples were capsule, powder, tablet, and soft-gel (Table S3). However, herbal supplements may have the complex matrix that can potentially interfere with the determination of target compounds and cause matrix effect in mass analysis [10]. The matrix effect should be evaluated for further method development and in for a variety plant food and herbal supplement samples.

Consider adding representative MS chromatograms on analyzed dietary samples to show the peak detection quality.

Authors: We have added the example of chromatogram and mass spectra in Figure S2.

Round 2

Reviewer 1 Report

The revised manuscript can be accepted for publication as an article in Molecules.